# Triggering the 2022 eruption of Mauna Loa

Kendra J. Lynn [1] ✉, Drew T. Downs[1], Frank A. Trusdell[1], Penny E. Wieser [2], Berenise Rangel[2], Baylee McDade [1], Alicia J. Hotovec-Ellis [3], Ninfa Bennington[1], Kyle R. Anderson [3], Dawn C. S. Ruth [3], Charlotte L. DeVitre[2], Andria P. Ellis [1], Patricia A. Nadeau[1], Laura Clor[4], Peter Kelly[4], Peter J. Dotray[1] & Jefferson C. Chang[1]

Distinguishing periods of intermittent unrest from the run-up to eruption is a major challenge at volcanoes around the globe. Comparing multidisciplinary monitoring data with mineral chemistry that records the physical and spatio-temporal evolution of magmas fundamentally advances our ability to forecast eruptions. The recent eruption of Mauna Loa, Earth's largest active volcano, provides a unique opportunity to differentiate unrest from run-up and improve forecasting of future eruptions. After decades of intermittent seismic and geodetic activity over 38 years of repose, Mauna Loa began erupting on 27 November 2022. Here we present a multidisciplinary synthesis that tracks the spatio-temporal evolution of precursory activity by integrating mineral and melt chemistry, fluid inclusion barometry, numerical modeling of mineral zoning, syn-eruptive gas plume measurements, the distribution and frequency of earthquake hypocenters, seismic velocity changes, and ground deformation. These diverse data indicate that the eruption occurred following a 2-month period of sustained magma intrusion from depths of 3–5 km up to 1–2 km beneath the summit caldera, providing a new model of the plumbing system at this very high threat volcano. Careful correlation of both the geochemistry and instrumental monitoring data improves our ability to distinguish unrest from the run-up to eruption by providing deeper understanding of the both the monitoring data and the magmatic system—an approach that could be applied at other volcanic systems worldwide.

Forecasting the timing of volcanic eruptions with certainty remains a critical challenge in volcanology[1,2]. Distinguishing between periods of unrest that wax and wane versus unrest that heralds an eruption within days to months is complicated, as demonstrated by recent events at frequently active volcanoes like Sierra Negra[3] (Galápagos Islands), Cumbre Vieja[4] (Canary Islands), Fuego[5] (Guatemala), and Nyiragongo[6] (Democratic Republic of the Congo). Traditionally, real-time monitoring of volcanic unrest has been used to infer magma storage and transport during the run-up to eruption. With improved analytical capabilities and the development of diffusion chronometry techniques (the use of chemical zoning in minerals to extract time-scales of magmatic processes), mineral chronologies can now link where and when magmas moved prior to eruption with monitoring signals measured at the surface. Careful correlation of both geochemical and monitoring datasets can improve interpretations of unrest versus run-up by providing deeper understanding of both the monitoring data and the magmatic system. Despite the potential for these powerful studies to advance our forecasting abilities, few multidisciplinary syntheses at active volcanic systems that leverage mineral chronometry exist.

[1]U.S. Geological Survey, Hawaiian Volcano Observatory, Hilo, HI, USA. [2]Department of Earth and Planetary Science, University of California, Berkeley, Berkeley, CA, USA. [3]U.S. Geological Survey, California Volcano Observatory, Moffett Field, CA, USA. [4]U.S. Geological Survey, Cascades Volcano Observatory, Vancouver, WA, USA. ✉e-mail: klynn@usgs.gov

Of the 585 volcanic systems on Earth with an eruption from 1500 to 2024 CE[7], only 9% (n = 56) have published diffusion chronometry studies (Fig. 1A) and only four (Kīlauea, Mount Ruapehu, Piton de la Fournaise, and Mount Etna) have been studied using multiparametric approaches where diffusion chronometry results can be correlated with four or more volcano monitoring data types (Supplementary Data 1). These combined studies enable a better understanding of eruptions at volcanoes that lack geophysical and gas monitoring data, and they have the potential to contribute to better interpretations of real-time monitoring signals and thus more accurate forecasting of eruptions globally[8]. This is especially true at systems where the most recent eruption pre-dates the development of a modern monitoring network, as was the case at Mauna Loa (Island of Hawaiʻi) in 2022.

Mauna Loa is the largest active volcano on Earth and has erupted 34 times since 1843 CE, posing significant hazards to island communities located on its flanks where voluminous and fast-moving lava flows can reach population centers in as little as a few hours[9]. Understanding how, when, and where magmas transit through the magmatic system is therefore of utmost importance for preparing for eruptions. Before 2022, Mauna Loa had last erupted in 1984, which was prior to the installation of the modern monitoring network. In 2002–2022, Mauna Loa exhibited increased rates of seismicity and ground deformation (Fig. 1B) that ultimately culminated in the most recent eruption[10]. At the time, however, it was difficult to assess if any particular pulse of unrest would lead to eruption in the near term. The Volcano Alert Levels and Aviation Color Codes were changed between "Normal/Green" and "Advisory/Yellow" several times during this period as the intensity of unrest waxed and waned (Fig. 1B).

Meanwhile, detailed geochemical analysis and diffusion modeling of Mauna Loa mineral zoning had not been performed for any eruption in the past 200 years, and only a few such studies had been conducted on Mauna Loa eruptions of any age[11,12]. In addition, the majority of Mauna Loa eruptions in the past 200 years lacked instrumental

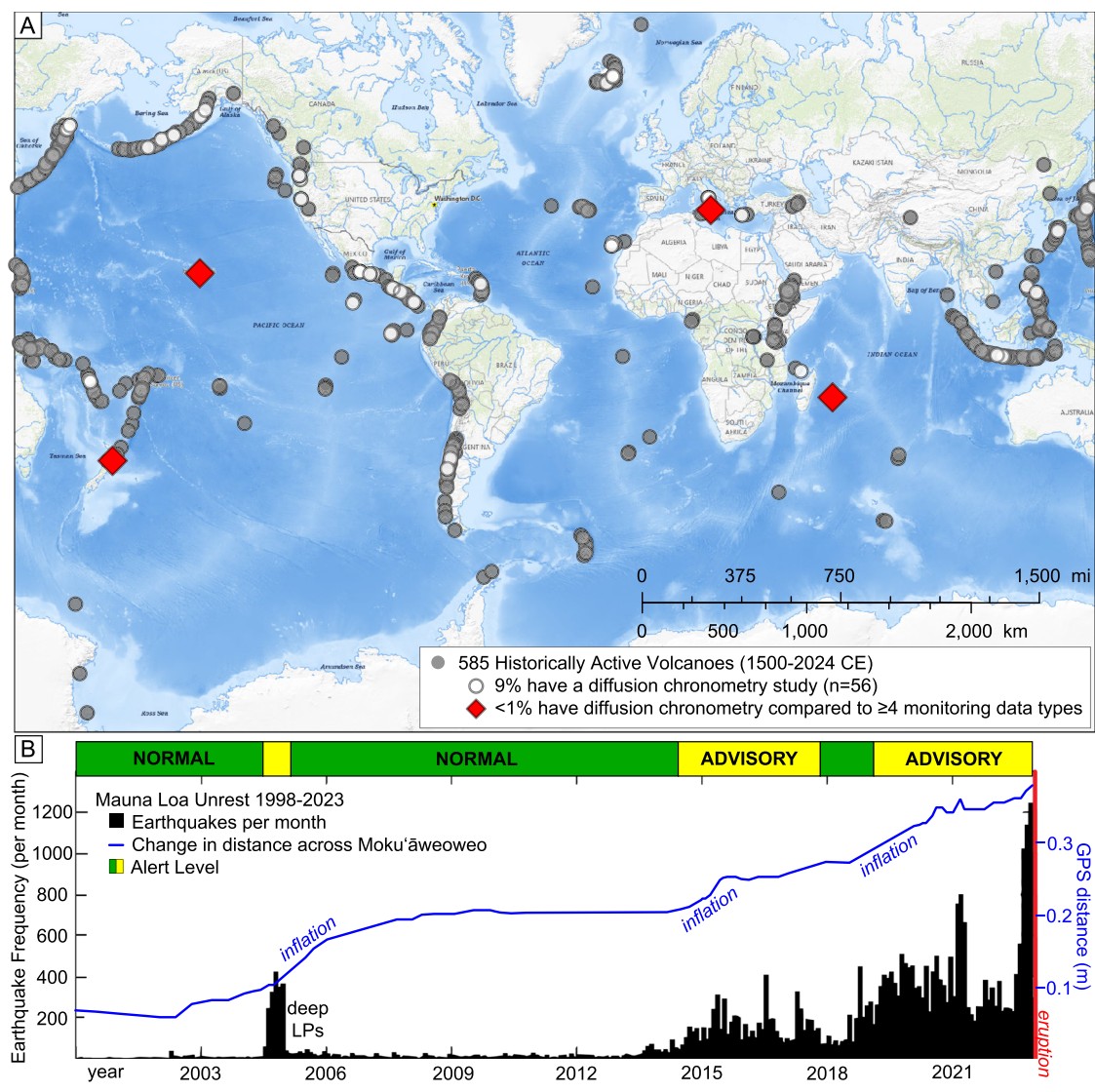

**Fig. 1 | Summary of volcanoes with diffusion studies and Mauna Loa unrest 2000–2022. A** Global distribution of volcanoes with diffusion studies (white circles, *n* = 56; Supplemental Bibliography) conducted on historically active and volcanoes with an eruption 1500–2024 CE (gray circles, *n* = 585[7]). Among those volcanoes with diffusion studies, only four (Kīlauea, Mount Ruapehu, Mount Etna, and Piton de la Fournaise) have multidisciplinary syntheses where diffusion timescales were compared to three or more types of monitoring data (see Supplementary Material). Basemap from the USGS National Map[77] **B** Long-term unrest at Mauna Loa volcano, Island of Hawaiʻi (1998–2023) shown by earthquake frequency counts (per month; black bins) and increasing distance between GPS sites across the summit caldera Mokuʻāweoweo (blue line; MOKP-MLSP in Fig. S1). Volcano Alert Levels and Aviation Color Code designations are shown as colored bar at top of figure. LPs are long-period earthquakes.

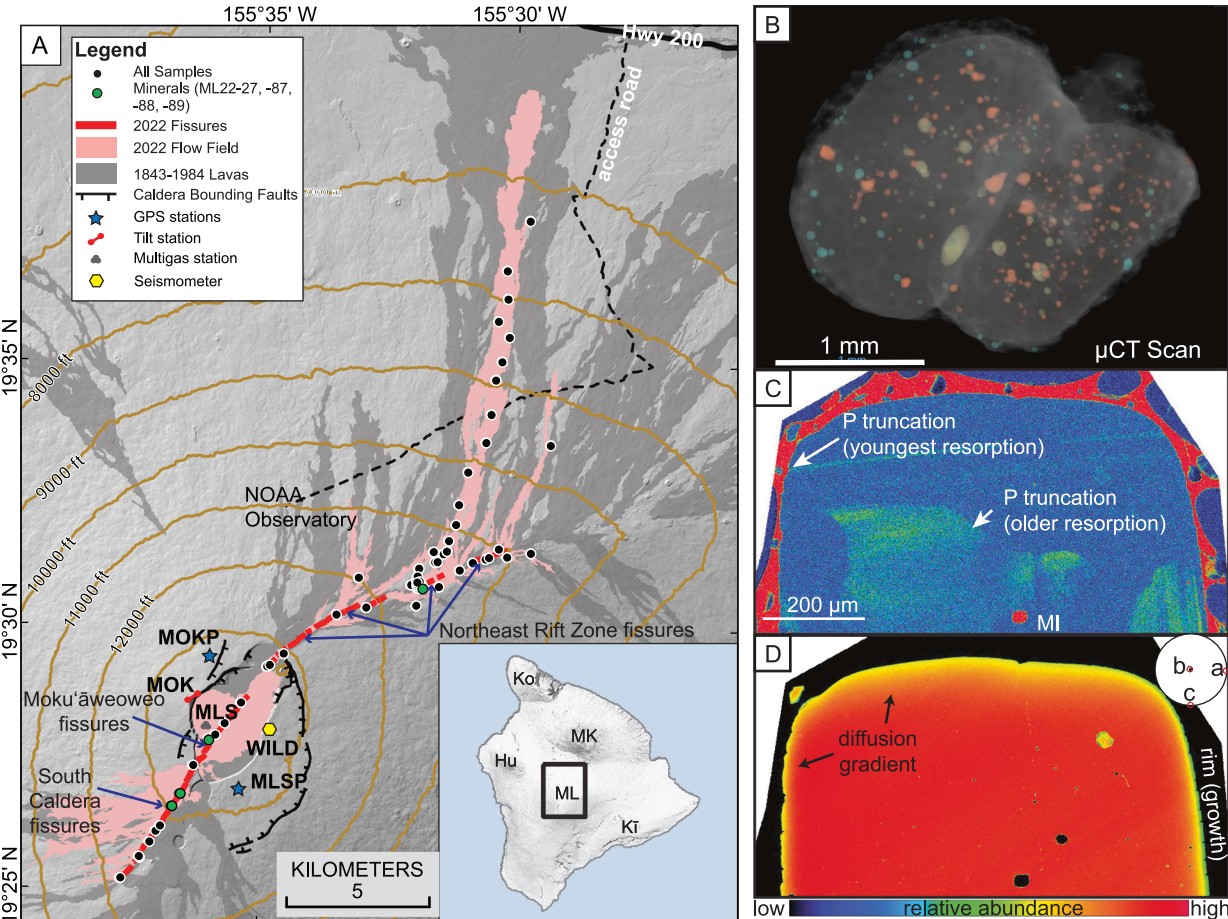

**Fig. 2 | Overview of Mauna Loa 2022 eruption and olivine crystal petrography.**
**A** Map of Mauna Loa summit and upper Northeast Rift Zone (Island of Hawai'i on inset map) with 2022 lava flow field[78] (pink), lava flows since CE 1843[79,80] (dark gray), 1000 ft (~305 m) contours (brown lines), samples (black dots; green for large-volume tephra samples used for mineral chemistry), and monitoring network stations (various symbols; see Fig. S1 for wider view that includes all GPS stations in this study). Basemap generated from a 2005 NOAA digital terrain model[81] and a 1983 USGS digital elevation model[82]. Caldera bounding faults are denoted as black lines with teeth pointing to the downthrown side and 2022 eruptive fissures are denoted with red lines. The 2022 flows crossed the NOAA Observatory access road (dashed black line) and came within ~3 km of Highway 200 (Daniel K. Inouye Highway, "Saddle Road"). **B** 3-D X-ray computed tomography scan (μCT scan) of an olivine from sample ML22-88 shows the rounded morphology typical of phenocrysts from this eruption and inclusions of melt (yellow), Cr-spinels (red), and fluids (blue). **C** Phosphorus and **D** magnesium X-ray element maps of an olivine crystal. Sharp truncations in phosphorus zoning record at least two dissolution events. Patchy zones of truncated high-phosphorus in the phenocryst interior represent older resorption histories, whereas truncation of secondary phosphorus branches by the crystal-melt boundary record the youngest resorption event that is also evident in the rounded morphology of the phenocryst. Diffusion gradients in magnesium show obvious differences along the *a*- vs. *c*-axes (shown in lower-hemisphere stereonet inset in **C**), a result of diffusion anisotropy. Part of the crystal rim in the *a*-axis direction also shows subtle skeletal rim texture, evidence of late-stage crystal growth.

monitoring data, which limited our understanding of periods of unrest and the likelihood and timing of the next eruption.

Here we integrate mineral and melt chemistry, fluid inclusion barometry, numerical modeling of mineral zoning, syn-eruptive gas plume measurements, distribution and frequency of earthquake hypocenters, seismic velocity changes, and ground deformation to show that a 2-month-long sustained intrusion of magma into shallow storage preceded Mauna Loa's 2022 eruption. This multidisciplinary synthesis integrates six monitoring data types (Supplementary Information), mineral and glass chemistry, diffusion chronometry, and fluid inclusion barometry—a successful example of combining research and monitoring practices to improve future forecasting capability at a well-monitored, active volcano.

## Results and discussion
### Overview of the 2022 eruption
After 38 years of repose, Mauna Loa began erupting at ~11:21 p.m. HST on 27 November 2022. The eruption commenced after ~30 min of continuous seismic tremor[13] with fissures opening in the southwest corner of the summit caldera (Moku'āweoweo) that propagated to the northeast, crossing the summit caldera, and farther to the southwest extending into the southern caldera region toward the Southwest Rift Zone (Figs. 2A and S1). These fissures were active for ~6 h before summit activity waned and a dike propagated into the Northeast Rift Zone in the early hours of 28 November. Four Northeast Rift Zone fissures were active between 3755 and 3365 masl (Fig. 2A), but activity localized to a single active vent by 2 December. The eruption diminished in intensity abruptly on 8 December and ended on 10 December after 13 days of activity.

During the eruption the U.S. Geological Survey (USGS) collected lava and tephra samples to characterize lava chemistry. The erupted lavas were chemically homogeneous, and their composition was more evolved than any other Mauna Loa eruption over the past 200 years (whole-rock average MgO 6.2 wt%[14] compared to 1984 average MgO 6.7 wt%[15]). The lava composition is consistent with "reservoir magma" for Mauna Loa[16], which is defined as having a composition "perched" at the low-MgO end of olivine control. This composition reflects storage in a long-lived, steady-state reservoir at a depth of ~3 km[16,17]. Samples

erupted in 2022 initially appeared to be aphyric, but large-volume (>1 m³) tephra samples were crushed, sieved, and picked, resulting in discovery of extremely rare olivine ($n = 64$) and enstatite ($n = 2$) macrocrysts >0.5 mm in size (Figs. 2B and S2) in four samples collected from the south caldera, Mokuʻāweoweo, and the Northeast Rift Zone (Supplementary Information). Olivine crystals predominantly have core forsterite (Fo; $Mg/(Mg + Fe) \times 100$) contents of >$Fo_{87}$ and all are normally zoned toward $Fo_{78-82}$ rims (Fig. S3). Olivine cores and rims are not in equilibrium with the erupted glass (MgO 4.6–6.2 wt%, Mg# 40–53) whereas enstatite cores are in equilibrium and rims are not (Fig. S4). Core-to-rim Fe–Mg zoning was modeled using 59 olivine and 2 enstatite macrocrysts, yielding timescales of diffusive re-equilibration prior to eruption that record the timing of magma migration and mixing in the plumbing system (Supplementary Data 1).

### Eruption triggered by 2-month-long sustained intrusion

Timescales of olivine Fe–Mg diffusive re-equilibration have been converted to "days prior to eruption" (e.g., $t = $ [calculated diffusion time] – [the day of the eruption on which the sample reached the surface]) so that model results from samples erupted on different days can be compared (Fig. 3). Around 81% of the diffusion dataset records re-equilibration timescales of ≤70 days (10 weeks) prior to eruption, corresponding to mid-to-late September 2022. Within the error of the diffusion models, 90% of the olivine crystals record timescales of ≤70 days. The two enstatite models return similar diffusion timescales.

The mineral chronometry correlates remarkably well with trends recorded in geophysical monitoring datasets, including the onset of increased frequency of summit earthquakes[18], changing patterns of earthquake hypocenters, changes in seismic velocities across the summit region[19], inflation recorded by continuous GPS line length changes across Mokuʻāweoweo[20–23], and inflationary tilt signals caused by magma emplacement recorded by the MOK summit borehole tiltmeter[24] for the first time since its installation[10] in 1999.

The MOK tilt signal was particularly significant[10]. Ten weeks prior to eruption, MOK recorded increasing rates of ground tilt away from the summit (Fig. 3A) coincident with increasing rates of extension between continuous GPS stations MLSP and MOKP (Fig. 3B), which are situated on opposite sides of Mokuʻāweoweo (Fig. 2A) and are sensitive to shallow subsurface changes. The change in tilt rate was only recorded at the MOK station, which can be interpreted as a shallowing of the deformation source. Although MLSP-MOKP line length changes occurred in previous episodes of unrest (Fig. 1B), no corresponding MOK tilt excursions related to magmatic processes were observed[10]. A decrease in seismic velocities (Fig. 3A) also occurred 10 weeks prior to eruption predominantly at summit seismic stations, consistent with dilatational strain due to shallow inflation[25,26]. These changes were contemporaneous with a significant increase in rates of seismicity at Mauna Loa's summit (Fig. 3C), with the highest rates of seismicity seen in 2022[18,27].

The characteristics of volcano-wide seismicity also changed coincident with the onset of MOK tilt. Prior to late September 2022, earthquake swarms alternated between faults[28] located at 6–11 km depth beneath the northwest flank (NWF, Fig. 3) and beneath Mokuʻāweoweo (Sum; Figs. 3 and S5, S6) 1–5 km below the surface. This alternating pattern began shortly after a noticeable increase in line length between continuous GPS stations ALEP and AINP (Fig. 3B), which are located on the lower flanks of Mauna Loa (Fig. S1), suggesting increased, complex pressure changes from magma emplacement in a dike-like source[29]. Swarms may be indicative of stress changes at intermediate depths below the summit and could record intrusive activity in the deeper portions of the reservoir system. In the 10 weeks prior to eruption, earthquake swarms instead occurred simultaneously in both locations (Figs. S5 and S6) at increased rates (Fig. 3C), yet with largely unchanged focal mechanisms[18]. While the causes of alternating versus contemporaneous earthquakes in the

NWF and summit regions are unexplored, the change to synchronous activity in 2022 correlates with changes in all other monitoring datasets presented here and may reflect an important change in the magmatic system.

This multidisciplinary synthesis of seismic, geodetic, and mineral data strongly suggests that a 2-month-long sustained magmatic intrusion[10] to the shallowest portions of the reservoir system ultimately triggered the 2022 eruption of Mauna Loa. An extended period of magmatic recharge to shallow levels is also supported by syn-eruptive multi-GAS measurements of the eruptive plume, which had a $CO_2/SO_2$ ratio of $0.28 \pm 0.05$ (Supplementary Data 1). These results are indicative of two-stage degassing comprising deep $CO_2$ loss and re-equilibration at shallower levels, similar to classic "Type 2" gas[30] at Kīlauea. The low ratio likely precludes the single-stage degassing that would occur with the rapid ascent of magma and gas from mantle depths.

### Records of unrest: years-to-decades-long history of recharge and intrusion

Several periods of unrest occurred at Mauna Loa following the 1984 eruption, including increased rates of seismicity and inflation in 2002–2005, 2014–2017, and again at the beginning of March 2021[10,28] (Fig. 1B). Relocated long-period earthquake swarms for 2002–2005 suggested magma migration from depths of ~45 km[28], which were followed by inflation consistent with magma entering the volcanic edifice[31–33]. Geodetic modeling of 2002–2005 inflation indicated that magma generated in the mantle ascended into the deep and intermediate regions of the plumbing system (>4 km), with very little material reaching shallower levels[29]. The shallowest seismicity was defined by earthquake swarms at 2–3 km depth beneath the southwest margin of Mokuʻāweoweo, capping an aseismic region interpreted to be the primary magma reservoir[28,32]. The seismic and geodetic data clearly indicated magma ascent through the reservoir system, but at the time there were no other constraints on reservoir depths or geometry based on petrological methods to corroborate inferred magma storage regions.

The record of older (pre-2022) recharge is not preserved in glass (Supplementary Data 1) and bulk-lava compositions[14] due to efficient mixing within storage areas that limits the compositional heterogeneity of the melt and swiftly erases the geochemical signal of less voluminous mafic melts that episodically intrude into more evolved, longer-residence magmas (as suggested for Kīlauea[34]). The 2022 olivine macrocrysts are the only discernible remnants of the deeper recharge by more primitive melts and offer additional insights into the years-to-decades-long history of unrest via a wide range of resorption and late-stage growth textures (Fig. 2C, D and Supplementary Data 1). Element X-ray maps of phosphorus, a slow diffusing element that preserves histories of crystal growth and dissolution[34,35], show patchy zones of truncated high-phosphorus in the high-Fo olivine interiors (Fig. 2C). In Kīlauea olivine, this texture has been interpreted to represent older resorption histories related to repeated recharge of magma into reservoirs that causes a complex multi-stage, pre-eruptive history of crystal dissolution and regrowth, all at high Fo[34,36]. Additionally, some 2022 olivine have subtle reverse zoning in their interiors (e.g., ML22-87 olivine 4 has a core of $Fo_{88.3}$ reversely zoned to $Fo_{88.6}$; Fig. S7) that cannot be accurately modeled for timescales but may represent some pre-2022 history of recharge, intrusion, and re-equilibration at high Fo.

### Redefining a model for the Mauna Loa plumbing system

Our multidisciplinary synthesis and olivine-hosted fluid inclusion barometry show that there are two sub-summit magma reservoirs. Classic interpretations of magma storage based on geophysical datasets supported a single, primary reservoir at 3–5 km beneath Mauna Loa's summit[37–39] (Fig. 4). Geodetic data modeled this storage zone as a

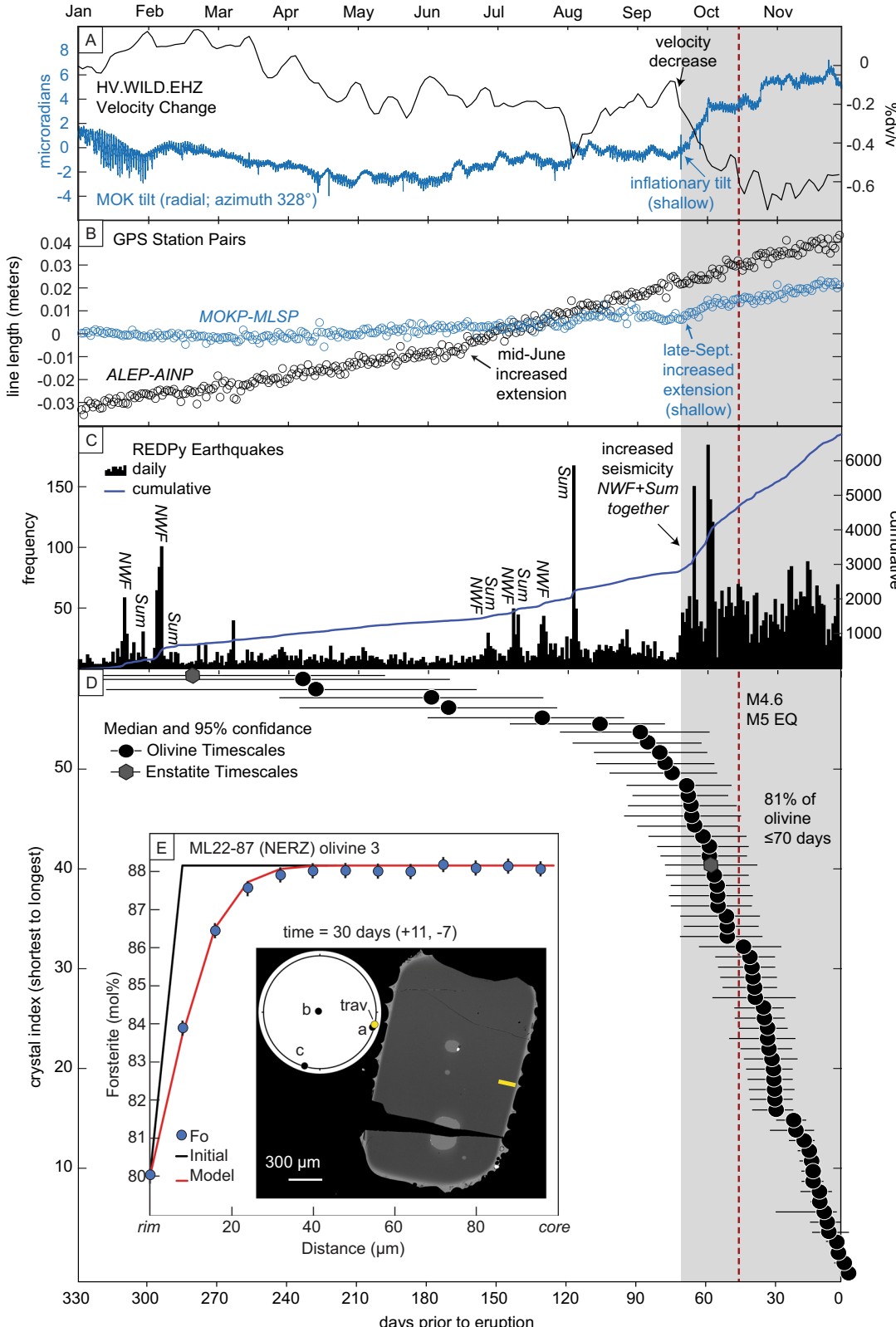

**Fig. 3 | Timeline of 2022 pre-eruptive unrest.** Multiparametric synthesis comparing **A** tilt[24] measured at station MOK (azimuth 328°) and change in seismic velocities at station WILD[19] (likely due to strain and crack opening from the same shallow inflation source(s) as the geodetic data), **B** change in line length between GPS stations MOKP and MLSP[20,21] and stations ALEP and AINP[22,23], **C** daily summit earthquake counts (black bins) and cumulative earthquake counts (blue line) derived from REDPy[18] and **D** calculated olivine and enstatite diffusion timescales with uncertainties. Station locations are shown in Fig. 1 (ALEP-AINP shown in Fig. S1). **E** Diffusion model example for an olivine from ML22-87 (Northeast Rift Zone). Electron backscatter diffraction data (EBSD) shown as lower-hemisphere projection confirms ideal orientation of olivine. Grayscale BSE image showing olivine section and electron microprobe traverse (yellow line). Forsterite profile (blue circles) shown with uncertainty (±0.2 mol% at 95% confidence), initial condition (black line), and model best fit (red line) to the EPMA data.

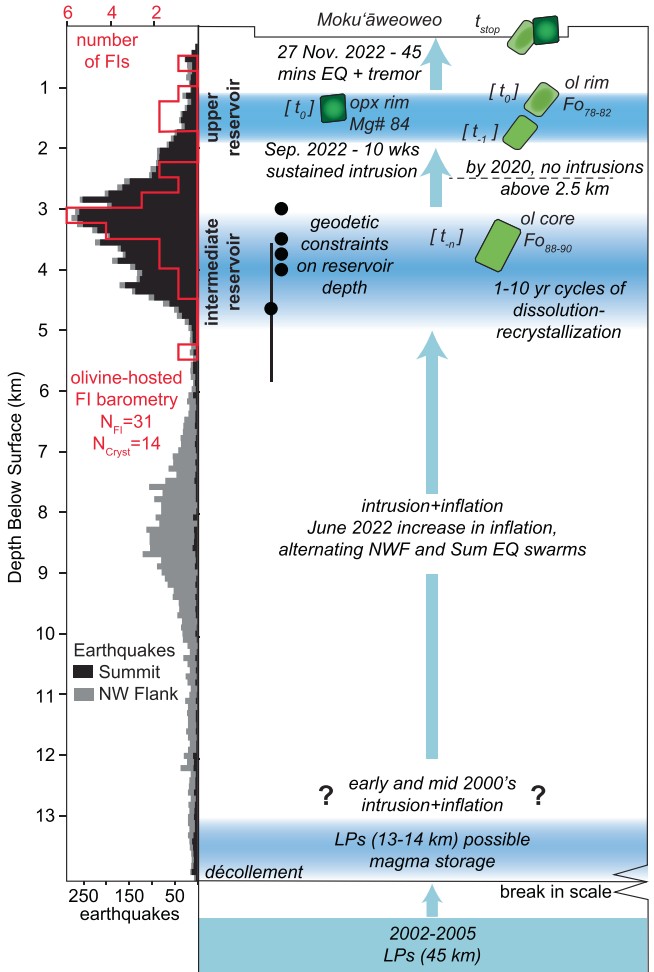

**Fig. 4 | Cartoon schematic of Mauna Loa's plumbing system.** Schematic integrates multidisciplinary evidence for magma storage regions (blue). Depth constraints from olivine-hosted fluid inclusion barometry compared to HVO/ComCat earthquake depths (for discussion of other earthquake catalogs, see Supplementary Material). Geodetic constraints on depths (black circles) of the intermediate reservoir come from ref. 29 (4.7 km depth with radius of 1.1 km), ref. 38 (3.5 km depth), ref. 39 (3.7 km depth), ref. 83 (4 km depth), ref. 84 (4 km depth), ref. 85 (3 km depth), and ref. 86 (13-14 km depth). Varugu and Amelung[40] constrained the 2.5 km depth limit of intrusions prior to 2020.

spherical source southeast of the caldera and a subvertical dike-like magma body bisecting the summit region[28,29,40]. Based on petrologic and geochemical data, Rhodes[16,17] suggested that a long-lived, large-volume reservoir with a low-MgO composition existed at 4 km depth, below the majority of earthquake hypocenters (Fig. 4). The reservoir was interpreted to be sufficiently large that it could repeatedly buffer signatures of mafic recharge.

The 2022 high-Fo olivine macrocrysts grew at depths consistent with the primary reservoir but could not have originated from the evolved reservoir magmas[16,17] at this depth. The median pressure calculated from the density of 31 $CO_2$-rich texturally primary fluid inclusions (FI) hosted in 14 olivine crystals using the $H_2O$–$CO_2$ equation of state[41] is -0.8 kbar (16th–84th quantiles = 0.4–0.95 kbar). Using an adapted density-depth model[41,42] this corresponds to depths of 3.46 km (1.8–4.1 km). There is a second cluster of 6 FI at depths of 0.6–1.8 km—these lower-density fluid inclusions tend to be located closer to the rims of crystals or in the core of smaller crystals (200–300 μm, see Supplementary Information) and may have been trapped during magma ascent or stalling in the shallower reservoir.

The FI barometry is likely not influenced by significant re-equilibration at the pressures and timescales in this study (Supplementary Information; Fig. S8) and instead we interpret that they reflect the depths at which crystal growth occurred.

The FI data indicate storage regions primarily at 1–2 and 3–5 km, which we call the upper reservoir and intermediate reservoir, respectively.

Although the olivine and enstatite are likely not phenocrysts (genetically related to their carrier liquids), the fluid inclusion data still document the dominant magma storage areas. The depth of the upper reservoir at 1–2 km (Figs. 4 and S9) redefines the Rhodes[16,17] interpretation of reservoir magmas. We suggest the reversely zoned enstatite macrocrysts with evolved core Mg# 78–79 grew at 1–2 km in equilibrium with the reservoir magmas in the upper reservoir and became reversely zoned up to Mg# 81–84 (out of equilibrium; Fig. S4) when mafic recharge magma intruded from the intermediate reservoir.

In late September 2022, the MOK borehole tiltmeter measured, for the first time, a clear inflationary signal related to magmatic intrusion into this shallowest part of the system[10]. Geophysical modeling[40] shows that prior to 2020, the shallowest portions of the inflating sub-summit magma body remained deeper than -2.5 km beneath the summit (Fig. 4). This is corroborated by timescales derived from zonation preserved in olivine and enstatite macrocrysts. All this evidence implies that the upper reservoir may only be ephemerally connected to the rest of the plumbing system through the subvertical dike-like magma body that bisects the summit caldera[40].

Upon introduction into the upper reservoir, olivine macrocrysts underwent resorption (i.e., dissolved), which was quickly arrested by diffusion and/or crystal growth, and the resulting timescales reflect storage after that intrusion. Olivine dissolves in a basaltic liquid that has a Mg# below the value dictated by the olivine-liquid Fe–Mg exchange equilibrium (for a given temperature, pressure, and oxygen fugacity), and it precipitates when Mg# of the liquid exceeds the exchange equilibrium value[43]. Truncation of phosphorus-rich zones by the olivine exterior (i.e., crystal-melt boundary; Fig. 2C) record the youngest resorption event, which is also evident in the rounded morphology shown in backscatter electron images of olivine macrocrysts (Supplementary Data 1) and 3-D scans of crystals using high-resolution X-ray computed tomography (Fig. 2B and Supplementary Videos).

High-Fo olivine can dissolve rapidly (a few hundreds of μm/hour) in more evolved liquids and diffusive exchange of Fe–Mg between crystal and liquid can also occur rapidly at magmatic conditions[34]. Intrusion of high-Fo olivine crystals into the more evolved "reservoir magma"[16,17] in September 2022 thus likely caused the final dissolution of the olivine that was quickly arrested as diffusive re-equilibration generated a lower Fo rim (78-80) and "armored" the crystals starting in late September 2022 (Fig. 2B–D). These textures, coupled with the scarcity of olivine crystals in the 2022 eruption products, could indicate that much of the olivine cargo associated with older (pre-2022) intrusions have completely dissolved upon introduction to more evolved magmas prior to eruption. Reservoir lavas at or past the low-MgO end of olivine control[16,17] would be too evolved to form low-Fo olivine cores and likely produced the low-Mg# enstatite cores.

## Why did Mauna Loa finally erupt in 2022?

What changed in 2022 that led to an eruption after years-to-decades of intrusion and unrest? We propose that the trigger was migration of sufficient magma into the upper reservoir until stresses exceeded the point of failure. Over the past 2 decades, a general long-term correlation between deformation and seismicity rates was observed (Fig. 1B), but with no obvious relation between them over shorter periods of time[28] and no coincident changes in MOK borehole tilt measurements. In contrast, simultaneous changes in earthquake frequency, geographic patterns of hypocenter locations (flank versus summit),

slowing of seismic velocities, tilt signals that suggest a shallow source, and increased inflation rate measured by GPS all indicated magma intrusion to shallower portions of the magmatic system than during previous periods of unrest. The key geophysical monitoring signal distinguishing between unrest and the final run-up to eruption at Mauna Loa is therefore likely the change in MOK tilt[10] and seismic velocities signaling the migration of magma to the very shallowest portions of the reservoir system at 1–2 km depth beneath the summit.

The synthesis of monitoring data and petrologic research presented here suggests that future eruptions might be preceded by tiltmeter and GPS signals with contemporaneous summit and flank earthquake swarms that all reflect magma accumulation in the upper reservoir. These combined indicators may improve our ability to forecast an eruption with more lead time than the minutes to hours of intense pre-eruptive seismicity typical for Mauna Loa eruptions.

At other historically active volcanic systems, careful correlation of geochemical and monitoring datasets can similarly improve interpretations of unrest by providing a deeper understanding of both the monitoring data and magmatic system. Diffusion chronometry studies have only been conducted at ~9% of the world's volcanoes with an eruption since 1500 CE (Fig. 1) but they have the potential to strengthen forecasting of eruptive activity. Mineral chronologies from past eruptions add to and enhance historical and geological records that inform how often a volcano tends to erupt and support long-term pattern recognition[2], in addition to identifying eruption triggers as was done retrospectively in this case at Mauna Loa. Although studies of erupted samples can only be carried out post-eruption, they can improve scientists' ability to forecast future eruptions by providing otherwise unobtainable insights into pre-eruptive magma storage and transport. With multiparametric diffusion chronometry syntheses on multiple volcanic systems globally, we may begin to elucidate common themes that could be applied to many of Earth's volcanoes.

## Methods

### Sample preparation
Large-volume (>1 m³) tephra samples were crushed and sieved into 1–2 and 0.5–1 mm size fractions. Mineral separates were picked from both fractions, with the majority of the macrocrysts found in the 1–2 mm size fraction. For diffusion work, whole macrocrysts (olivine and enstatite) and crystal clusters were individually oriented down either the *a*- or *b*-axes and sectioned through their cores[44], yielding nearly ideal 2-D sections despite their resorption textures. This technique greatly reduces or eliminates uncertainties in diffusion modeling related to off-center and oblique sectioning. The efficacy of this methodology is shown with EBSD confirmation of crystal orientations, which were used in diffusion modeling. Crystals were mounted in 2.5 cm epoxy rounds and polished to 0.3 μm.

### Electron probe micro-analysis (EPMA)
Olivine, enstatite, and glasses were measured using routine procedures using a five spectrometer JEOL Hyperprobe JXA-8530F+ at the USGS Lab in Menlo Park, California (see Supplementary Material for details and precision). Secondary standards were routinely measured to assess data quality and monitor for drift. Basaltic glass compositions for Mauna Loa eruptions (CE 2022, 1950, 1942, 1907, 1887, 1881, 1868, and 1852) were analyzed to assess an Fe–Mg equilibrium temperature for a given olivine Fo content (further details below). For all EPMA analyses (Supplementary Data 1), X-ray intensities were converted to concentrations using standard ZAF corrections (based on atomic number (Z), absorption (A), and fluorescence excitation (F))[45]. Analyses with totals <99.0 wt% or >101 wt% were rejected.

Olivine X-ray element maps of P (PETL), and Mg (TAP) were collected using wavelength-dispersive spectrometry on the JEOL Hyperprobe JXA-8530F at the USGS lab in Menlo Park, California, for two olivine crystals from the diffusion modeling suite. Analytical conditions were 15 keV, 200 nA, resolutions of 1 μm/pixel, and dwell times of 300 ms/pixel.

### High-resolution X-ray computed tomography scans
Two olivine macrocrysts were imaged in 3-D at the University of Texas High-Resolution X-ray Computed Tomography Facility (funded by the NSF Division of Earth Science Instrumentation and Facilities Program [NSF EAR-1762458] and NASA [80NSSC23K0199]). These macrocrysts were scanned using the 4× objective on a Zeiss Xradia 620 Versa. The X-ray source was set to 70 kV and 8.5 W with a 0.35 mm SiO$_2$ filter. A total of 2401 1.25 s projections were acquired over ±180° of rotation with no frame averaging. A source-object distance of 12.6 mm and a detector-object distance of 15 mm resulted in 3.09-μm resolution. The resulting data volumes were reconstructed as 16 bit TIFF stacks with a 0.3 beam-hardening correction and byte scaling of [−0.1, 1.4]. Scan duration was 1.73 h/sample. The 3-D visualizations of each crystal were rendered in Dragonfly (Object Research Systems v. 2023.2). The crystal was rendered semi-transparent, with false color applied to internal fluid inclusions (blue), melt inclusions (yellow), and Cr-spinel inclusions (red).

### Diffusion modeling
**Olivine.** Intrinsic environmental parameters required for the models are relatively well-constrained for Mauna Loa. The likely temperature at which diffusive re-equilibration occurred was determined using the Shea et al.[46] thermometer for Hawaiian tholeiitic basalts and a compilation of Mauna Loa eruption glasses from the past 200 years (this study for CE 2022, 1950, 1942, 1907, 1887, 1881, 1868, and 1852, ref. [47] for CE 1859, ref. [48] for CE 1950) to assess an Fe−Mg equilibrium temperature for a given olivine Fo content. Equilibrium between olivine and glass was assessed using $K_{D,FeMg}^{ol-melt} = 0.335 \pm 0.01$[46] and a $Fe^{3+}/\Sigma Fe = 0.16$[49]. The diffusion model temperature is akin to the magma temperature for pre-eruptive storage. Oxygen fugacity ($fO_2$) was set relative to the quartz-fayalite-magnetite buffer at QFM + 0.4 (similar to Kīlauea[50]). The pressure was estimated using the approximate center of the proposed upper reservoir at 1–2 km depth (42 MPa), supported by fluid inclusion barometry results.

Compositional gradients in Fo were modeled using finite differences and the one-dimensional form of Fick's second law[51]:

$$\frac{\partial C}{\partial t} = D \frac{\partial^2 C}{\partial x^2}$$

where *C* is concentration, *t* is time (in seconds), *D* is the diffusion coefficient in m²/s, and *x* is distance (in meters). We used the concentration-dependent diffusion coefficient for Fe−Mg in olivine[52,53]:

$$D_{Fe-Mg} = 10^{-9.21} \left(\frac{fO_2}{10^{-7}}\right)^{\frac{1}{6}} 10^{3(X_{Fe}-0.1)} \exp\left(-\frac{201000 + (P - 10^5)(7 \times 10^{-6})}{RT}\right)$$

where $fO_2$ is the oxygen fugacity (in Pa), $XFe$ is the mol fraction of iron in the olivine, *P* is the pressure (in Pa), *R* is the ideal gas constant, and *T* is the temperature (in Kelvin). This expression is for diffusion along the *c*-axis, which is six times faster than diffusion along the *a*- or *b*-axes. To account for diffusion anisotropy, a modified diffusivity was calculated using the measured orientations of the principal crystallographic axes relative to the traverse direction (Costa and Chakraborty[54]):

$$D_{trav}^{Fe-Mg} = D_a^{Fe-Mg}(\cos\alpha)^2 + D_b^{Fe-Mg}(\cos\beta)^2 + D_c^{Fe-Mg}(\cos\gamma)^2$$

where *α*, *β*, and *γ* are the angles between the traverse and the *a*-, *b*-, and *c*-axes of the olivine crystal, respectively. These were calculated from EBSD-determined orientations (see Supplementary Information).

A Bayesian approach which utilizes data and independent prior information was used to probabilistically constrain olivine diffusion

timescales[55]. We parameterized by model parameter vector $m = [\Delta t, T, p, fO_2, \alpha, \beta, \gamma, x_c, Fo_r, Fo_c]$, where $\Delta t$ is the duration of diffusion, $T$ and $p$ are the magma's respective temperature and pressure, $fO_2$ is the oxygen fugacity, $\alpha$, $\beta$, and $\gamma$ specify crystal orientation parameters in 3D space (see Supplementary Information for electron backscatter diffraction and crystal orientation details), and $x_s$ is the distance from the rim of the transition between rim and core forsterite concentrations $Fo_r$ and $Fo_c$, respectively. For an initial condition, we used a step (Heaviside) function at $x_s$; boundary conditions fix rim and core compositions to the specified values. We fixed diffusion parameters[56], so do not account for uncertainty in these values. A consideration of the faster diffusion coefficient proposed by Shea et al.[57] is presented in the Supplementary Information (Fig. S10).

We assumed that uncertainties in measured Fo are normally distributed with a standard deviation of 0.1 mol% (0.2 mol% at 95% confidence; see Supplementary Data 1 for EPMA precision). We used additional (prior) information to inform the values of some model parameters. For T we used a normal (Gaussian) prior distribution centered on the laboratory-estimated value for each sample, with a standard deviation of 13°, and for $fO_2$ we used a normal prior distribution centered on the laboratory-estimated value for each sample, with relative uncertainty of ±10%. For $p$ we used a prior distribution that is uniform between 20 and 110 MPa with Gaussian tails (standard deviation of 3 MPa) above and below, roughly corresponding to depths of 0.7–4 km. We fixed crystal orientation parameters to laboratory-derived values (see Supplementary Data 1). For other parameters we used uniform distributions between broadly spaced upper and lower bounds.

For each compositional profile we carried out a Markov Chain Monte Carlo (MCMC) inversion to sample the posterior probability density function for model parameter vector **m** (as defined above, except estimating $\log_{10}(\Delta t)$ rather than $\Delta t$) using the affine invariant ensemble sampler[58], as implemented in the GWMCMC code for MATLAB (https://github.com/grinsted/gwmcmc) by Aslak Grinsted. We utilized 150 walkers that iterated for a total of five million points per olivine sample and discarded 20% of the points as burn-in. To avoid bias caused by MCMC chains that did not converge, MCMC sample points with a probability more than five orders of magnitude lower than the maximum probability were discarded; for most olivine samples this was at most a few percent. Median values and equal-tailed 95% posterior credible intervals (confidence bounds) were computed from the marginal distributions for each model parameter. Uncertainties should be considered minima as they do not account for model error. Our Bayesian diffusion chronometry software for MATLAB is freely available[55].

**Orthopyroxene.** Two enstatite macrocrysts (one from the Northeast Rift Zone and one from Mokuʻāweoweo) were modeled for diffusive re-equilibration following the methodology of refs. 59,60. Both crystals have lower Mg# cores (78), likely reflecting the 1–2 km upper reservoir environment, mantled by higher Mg# rims (81-83) that probably indicate mafic recharge to the upper reservoir. Profiles were collected along the *a*- or *b*-axes and modeled using the new Fe–Mg diffusion coefficient[61]. The results are very similar to those retrieved using the orientation-corrected diffusion coefficient[62]. For example, the Northeast Rift Zone enstatite timescales are 59 days using ref. 61 compared to 75 days using ref. 62. We measured the Al profile concurrently to help constrain the shape of the initial Mg# profile. We also assumed a constant boundary at the rim that does not change with time. Model inputs utilized the same $T$ (1175 °C), $P$ (42 MPa), and $fO_2$ (QFM + 0.4) applied to the majority of the olivine diffusion models and are akin to the upper reservoir between 1 and 2 km. We minimized the sum of squares deviations (SSD) between the model and the observed values and then calculated the $R^2$ value; the two model results each returned $R^2 = 1$. Enstatite timescale uncertainties were estimated at

approximately ±30% of the modeled value, which is largely based on thermometer uncertainty of ±13 °C[46].

## Olivine-hosted fluid inclusion barometry

Individual crystals were mounted in CrystalBond™ on glass slides and ground down to within ~50 μm of the fluid inclusion of interest for Raman analyses. Crystals mounted in epoxy for Fe–Mg diffusion work were also inspected for fluid inclusions and analyzed where present. All fluid inclusions appear to have been trapped during growth of the crystal (termed primary fluid inclusions). This is based on the fact they are often isolated in crystal cores or present within clear growth zones delineated by spinels and melt inclusions. No fluid inclusions were present in linear trails (secondary fluid inclusions).

Raman analyses were performed using a WiTEC Alpha 300R Raman Spectrometer in the Department of Earth and Planetary Science, UC Berkeley, following the methods of ref. 63. Briefly, the sample was heated to 37 °C using a Peltier plate, monitored by a type K thermocouple, and excited with a 532.046 nm green laser through a 50× or 100× objective. Laser powers were set using the TruPower system and kept as low as possible to avoid issues relating to laser heating[63]. For fluid inclusions mounted individually in crystalbond, laser powers of 12 mW were used. Many fluid inclusions within the olivine mounts for diffusion were located at greater depths below the surface, requiring higher laser powers to get sufficient signals (12–25 mW), particularly for those that were small enough that the 100× objective had to be used. Peaks were fit using Psuedovoigt curves in the open source Python3 package DiadFit[64] to calculate the distance between the two diad peaks (splitting). Measured splitting was corrected for instrument drift using repeated acquisitions with a Neon lamp every ~10–20 min[65] using the 1117–1447 cm line. Corrected splittings were converted into $CO_2$ densities using the instrument-specific densimeter of ref. 63

To convert $CO_2$ densities into pressures, an estimate of the entrapment temperature was required. To determine this temperature, we used the relation between olivine Fo content and temperature parameterized for Kīlauea[63]. This gave very similar results to the regression used for diffusion temperatures but extends up to the higher Fo contents seen in crystal cores (no measured Mauna Loa liquids are in equilibrium with such high Fo values). For the olivines that were mounted in CrystalBond™ prior to FI analyses, we ground down as close to the level of the FI as possible prior to mounting in epoxy, and then analyzed the host crystal close to the FI using the USGS EPMA. For the fluid inclusions in the epoxy mount used for diffusion chronometry, we used the EPMA profiles to estimate the Fo content, by comparing the position of the profile to the position of the fluid inclusions. For the small number where EPMA analyses were not performed, we allocated the mean temperature of the dataset (1295 °C).

We first calculated pressures using the measured $CO_2$ density and estimated temperature using the pure $CO_2$ EOS[66]. This initial pressure estimate allows calculation of the probable amount of $H_2O$ in the exsolved fluid ($XH_2O$), using trends between pressure and $XH_2O$ developed at Kīlauea[63,67] (Fig. S11). This is a reasonable assumption for Mauna Loa, given the similar $H_2O$ contents of melt inclusions (Fig. S11). After allocating each FI an $XH_2O$ value, we calculated a bulk density for the fluid at the time of trapping following ref. 68, assuming $H_2O$ was diffusively lost prior to measurement. This bulk density was then input into the $H_2O$–$CO_2$ equation of state[69]. We then performed one final iteration−recalculating $XH_2O$ for this new pressure, and then recalculating pressure. Further iteration steps did not change pressure in any meaningful way given uncertainty in $XH_2O$. We converted pressures into depths using the Kīlauea density-depth model[41,42].

As well as calculating a pressure for each fluid inclusion, we also performed a Monte Carlo simulation with $N = 500$ values randomly drawn from distributions of the following input parameters.

- $CO_2$ density: normally distributed prior distribution combined errors from peak fitting, instrument drift, and the densimeter parameterization.
- Temperature: normally distributed prior distribution is centered on value calculated from the Fo value, with a normally distributed 1 sigma error of +50 K for fluid inclusions with a nearby probe measurement, and +100 K for those using the mean Fo content of the dataset (this is a generous error, as the standard deviation of measured Fo contents results in an error of 17 K).
- $XH_2O$: normally distributed prior distribution using sigma calculated from half the difference between the minimum and maximum parameterization of ref. 63.

The magnitudes of these propagated errors in terms of pressure and depth are shown in Fig. S12. The mean error is ±0.13 km, and 3.8 MPa.

## Multi-GAS measurements

Gas plume compositions ($H_2O$, $CO_2$, $SO_2$, $H_2S$) were measured in situ using a helicopter-borne USGS multi-GAS (multiple Gas Analyzer System) instrument[70,71]. The multi-GAS included an integrated GPS receiver (Garmin GPS 18 × LVC), a non-dispersive infrared $CO_2$ and $H_2O$ analyzer (LI-COR, Inc. LI-840A, 0–5000 ppm for $CO_2$, 0–80 parts per thousand for $H_2O$), and electrochemical $SO_2$ (City Technology, Ltd., T3ST/F, 0–200 ppm) and $H_2S$ sensors (City Technology, Ltd., T3H, 0–100 ppm). All data were logged at 1 Hz to a datalogger (Campbell Scientific, CR1000). The $H_2O$ and $CO_2$ data were automatically compensated for pressure and temperature. The manufacturer's empirical pressure correction was applied to the $SO_2$ sensor, and an ideal-gas-type pressure correction was made to the raw $H_2S$ sensor data. The instrument's $CO_2$, $SO_2$, and $H_2S$ sensor responses were verified after the flight with gas standards ($CO_2 = 1500$ ppm, $SO_2 = 10$ ppm, $H_2S = 10$ ppm) and were within 7% for $CO_2$, 11% for $SO_2$, and 30% for $H_2S$. Based on these data, the total analytical error estimate for the $CO_2/SO_2$ ratio was <20%. The $H_2S$ sensor was also determined to be ~12% cross-sensitive to $SO_2$ gas.

The helicopter intercepted the eruptive fissure gas plume on the first full day of the eruption (November 28, 2022) from 9:40:06 a.m. HST to 9:41:47 a.m. HST for a total data collection interval of ~1.5 min (Supplementary Data 1) at an altitude of 3858 m. Peak instantaneous (1 Hz) mixing ratios up to 6 ppmv $CO_2$ above atmospheric background ($\Delta CO_2$), and 21 ppmv $\Delta SO_2$ were recorded. Water vapor was not resolved above atmospheric background and $H_2S$ was not detected. The molar $CO_2/SO_2$ ratio ($0.28 \pm 0.05$, $r^2 = 0.58$, $n = 102$) was calculated from the time-series data based on the slope of the linear regression line for the pair of species. Further details of the collection and processing of multi-GAS data can be found in ref. 72.

## Seismic velocity change measurements

The timeseries of relative seismic velocity changes presented here is derived from coda wave interferometry of repeating earthquakes, following the methodology of ref. 25 and using the open source CWIRE software package[25,26]. The source catalog of earthquakes is the REDPy catalog[18,27] (Supplementary Material). The coda of each pair of earthquakes within a family as recorded on a single channel of seismic data are analyzed for relative compression or stretching due to a change in the seismic velocity structure between the times of the earthquakes' occurrence. All pairs of measurements are combined in a regularized least-squares inversion to find the continuous timeseries of velocity change that best fits the pairwise measurements. This process is repeated for all available channels of seismic data from seismic stations in the vicinity of Mauna Loa. Inversion results for all channels and at a variety of temporal resolutions are available in ref. 19, although we present only data from the vertical component (EHZ) at the permanent short period station WILD for simplicity.

## Earthquake catalogs

We queried all catalogs for located earthquakes of all magnitudes and depths occurring between the latitudes of 19.35° N and 19.6° N, and longitudes of 155.75° W and 155.52° W. The hypocentral depths in this catalog are reported in km below sea level; we adjust the depths to reference the surface by adding 4.17 km, the elevation of Mauna Loa's summit. For details specific to each catalog used (HVO/ComCat[73], REDPy[18], Matoza[74], Wilding[75]) and definition of "Summit" and "Northwest Flank" earthquake types, see Supplementary Material. For the more temporally complete REDPy catalog[18], events with no location or that occur outside this box are labeled separately.

## Data availability

Continuous GPS data are available through the GAGE/EarthScope Consortium (https://iris.edu); earthquake waveform data are available via SAGE/EarthScope Consortium (https://iris.edu); earthquake hypocenters are available via the Advanced National Seismic System's Comprehensive Catalog[73] (https://earthquake.usgs.gov/data/comcat/); the remaining data in this manuscript are available as Supplementary Tables and are available in the EarthChem Library[76] (https://doi.org/10.1594/IEDA/113447) or references cited.

## Code availability

Baydiff used for diffusion modeling error propagation is available at ref. 55.

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

## Acknowledgements

The multi-agency response to the Mauna Loa 2022 eruption was supported by staff at the U.S. Geological Survey's Hawaiian Volcano Observatory and Volcano Science Center, Hawai'i Volcanoes National Park, Hawa'i County Civil Defense, and the University of Hawai'i at Hilo. Samples were collected with permission from Hawai'i Volcanoes National Park in areas that were closed to the public during the eruption (permit HAVO-2021-SCI-0048). Any use of trade, firm, or product names is for descriptive purposes only and does not imply endorsement by the U.S. Government. We thank H. Lowers (USGS Denver) for assistance with EBSD analysis and J. Maisano (University of Texas) for collecting the CT scans. P.W., B.R., and C.D. acknowledge funding from NSF EAR 2217371, the Berkeley Rose Hills Innovator Program, the EPS Ramsden Fund, MPS scholars, and a SURF summer fellowship. The U.S. Geological Survey's

Volcano Hazards Program and the Additional Supplemental Appropriations for Disaster Relief Act of 2019 (P.L. 116-20) supported the work of K.J.L., D.T.D., F.A.T., B.M., A.J.H.-E., N.B., K.R.A., D.C.S.R., A.P.E., P.A.N., L.C., P.K., P.J.D., and J.C.C. The USGS thanks Michael Poland for providing the internal peer review.

## Author contributions

K.J.L. conceived the project, conducted field, laboratory, analytical, and modeling work, synthesized the interdisciplinary datasets, and wrote and revised the manuscript. D.T.D. conducted field, lab, and analytical support. F.A.T. collected all samples used in the study and supported interpretation and synthesis. P.E.W., B.R., and C.D. conducted fluid inclusion analysis and interpretation and EPMA analyses of olivine hosts for fluid inclusions. B.M. and D.C.S.R. assisted with sample preparation, EPMA data collection, and diffusion modeling. A.H.-E. provided seismic velocity change analysis, and with N.B. provided seismic interpretation. K.R.A. developed Baydiff and performed diffusion modeling error analysis. A.P.E. provided GPS network management, data collection, and deformation interpretation. P.N., L.C., and P.K. provided the multi-GAS instrument, field coordination, and subsequent data reduction and interpretation. P.J.D. and J.C.C. provided seismic network management and data collection. All coauthors contributed to the manuscript revision.

## Competing interests

The authors declare no competing interests.
