## [Peer Review File · Nature Communications]

Triggering the 2022 eruption of Mauna LoaREVIEWERS' COMMENTS

Reviewer #1 (Remarks to the Author):

In this manuscript, Lynn et al. present the results of a multi-disciplinary study centred on the 2022 eruption of Mauna Loa on Hawai'i. The authors combine mineral chemistry analysis with multiple geophysics datasets to better understand how to differentiate between non-eruptive unrest and pre-eruptive unrest at the volcano. The authors track precursory activity in 2022 through mineral and melt chemistry, fluid inclusion barometry, numerical modelling of mineral zoning, syn-eruptive gas plume measurements, earthquake locations, seismic velocity changes, and ground deformation. They find evidence for a sustained intrusion of magma into a shallow magma reservoir only 1-2 km depth beneath the summit. The authors conclude that careful integration of geochemical and geophysical datasets is key for distinguishing non-eruptive from pre-eruptive unrest.

Overall, I think this is a really important piece of work that clearly demonstrates the vital importance of integrating geophysical and geochemical datasets. While I don't have enough experience to carefully evaluate the geochemical analyses presented here, I do appreciate that a huge amount of work has come together to produce this manuscript. I do not find many issues with the manuscript, and the issues I do find can be easily resolved; I have listed my comments below. Once these comments have been addressed, I look forward to seeing this work publishing in Nature Communications.

Oliver Lamb
Te Pū Ao | GNS Science
o.lamb@gns.cri.nz

Major comments

1. The authors emphasise in multiple places that their work can help distinguish between non-eruptive and pre-eruptive activity but I'm not entirely sure the difference has been made explicitly clear. For example, did the authors compare the geophysical measurements between 2022 and earlier unrest periods, particularly in 2014? In fig. 1B, the inflation measured across the MOKP-MPLS in the 2014 unrest is rapid but I could not see anything in the text that explains whether this deformation had come from a shallow or deep source. Similarly, where were the earthquakes in the 2014 event located? If there is a clear distinction between the 2014 unrest and the 2022 pre-eruptive unrest, this would help support their conclusions (especially on line 273 to 280).

2. As a volcano-seismologist, I'm left with some questions about the seismic data that could easily be answered:

> What is the cause of the earthquakes in NW flank, as they sit between the deeper magma storage at 13 km and the shallower 'oma waena reservoir? I understand they originate from deep seated faults, as stated on line 147, but do you have an idea on what is causing them to rupture? Is it stress

from ascending magma or from the deeper reservoir?

> Do you have an idea of why we see relatively little seismicity associated with the 'oma luna reservoir, compared with the 'oma waena reservoir? Is it because the shallow seismicity was dominated by tremor which might not be accounted for in earthquake catalogues, or was there simply no detectable seismicity in this region? If the latter, is there a distinct difference in the geology around each reservoir that might explain this difference in seismic activity?

Alternatively, is it possible that the use of a barometer parameterised for Kilauea is not appropriate for Mauna Loa resulting in estimated reservoir depths that are too shallow? I don't really see this discussed anywhere in the manuscript. Please pardon my ignorance since I'm not really experienced in geochemical barometry techniques!

Minor comments

Line 52 – 54: Do you have references you can cite for each of these examples?

Line 62 – 63: I understand you've focused on diffusion chronometry studies because that's what this study is mostly focused on, but this seems a bit narrow. What about other studies that integrate geochemical with geophysical studies? For example, we did a study on Santiaguito volcano that integrated geophysical and geochemical measurements to shed light on why volcanic activity changed (Wallace et al., 2020). I would encourage the authors to expand the scope of their review of multi-disciplinary studies but I understand if this quickly becomes too much work to describe or reference.

Fig. 1B: What do the ticks in the lower panel refer to? They don't seem to line up with the year labels on the x-axis.

Line 102: I think you meant to refer to Figure 2A here, not Figure 1B.

Figure 3: From where on the volcano did the peaks seismicity 300-330 days prior to eruption come from? (panel C). There is also a typo in one of the labels here ('Increased siesmicity', should be 'increased seismicity').

Line 157: I think 'correlates' would be a better word to use than 'corresponds' here.

Line 279: Partially related to one of my major comments, but how do we detect magma ascending into the very shallowest portion of the reservoir system if there is no seismicity that comes from this region? Is it possible to detect this using the tilt signals alone? This is an important point to make as this kind of information might not be immediately available for volcano observatories to use.

References

Wallace, P.A., Lamb, O.D., De Angelis, S., Kendrick, J.E., Hornby, A.J., Díaz-Moreno, A., González, P.J., von Aulock, F.W., Lamur, A., Utley, J.E.P., Rietbrock, A., Chigna, G., Lavallée, Y., 2020. Integrated constraints on explosive eruption intensification at Santiaguito dome complex, Guatemala. *Earth and Planetary Science Letters* 536, 116139. <https://doi.org/10.1016/j.epsl.2020.116139>

Reviewer #2 (Remarks to the Author):

Review of "Triggering an eruption at Mauna Loa, Earth's largest volcano" by Lynn and colleagues

This ms presents a very detailed and comprehensive synthesis of multidisciplinary (both geochemical and geophysical) data associated with the most recent eruption (Nov 2022) at Mauna Loa, with the goal to advance the ability to forecast eruption. Such a multidisciplinary study is rare but critical for better understanding of eruption. It is important and of great interests to general readers. It should be published after moderate revision. I don't have any major issue with the manuscript, and all my comments are to ask for clarification. I encourage the authors to make the manuscript more approachable to non-experts.

Since I am a geochemist, my comments are related to mineral and melt inclusion chemistry. I leave the discussion on earthquakes to other reviewers.

1. Figure 3: Is timescale uncertainty of individual mineral considered in the histogram in Panel D? If possible, may I suggest replacing the histogram with a probability density function diagram, and taking into account the uncertainty of individual mineral?
2. You may also want to make the X-axis of the inset the same as your main figure. This allows a direct comparison with other parameter.
3. It is unclear how well (or bad) the diffusion time scales are correlated with other parameters. For example, there is a peak between 4-6 weeks. However, this peak seems not to be associated with any in Panels a-c.
4. Calculation of diffusion time scale:
 - a. A more detailed description is needed.
 - b. Have the authors used a 2D or 3D diffusion model?
 - c. The discussion about "olivine Fe-Mg diffusion coefficients" seems circular.
 - d. Some of the fitting may not reproduce the observation. For example, the first panel (ML22-ORI-1 ol1) does not show good fitting at the corner.
5. Figure 4: melt inclusion depth. I strongly suggest a more detailed description of how each individual melt inclusion depth is estimated, based on what parameter, and how (partial) degassing is excluded.
6. Figure S8 and associated discussion: Again, a more detailed description is suggested, so that non-experts can understand.
7. Raw data are needed for the correlation of olivine core-rim composition and melt inclusion depth shown in Figure 4. They may be in Tables S4 and S7; however, I expect to see the above mentioned correlation better illustrated using data.
8. Finally, the reference styles are different in main text and appendix. Is numbered reference system required for both?

Reviewer #3 (Remarks to the Author):

This is a fantastic multi-disciplinary effort to combine petrological methods with geophysical monitoring (seismic and ground deformation data) and gas geochemistry to constrain the drivers of the 2022 Mauna Loa eruption. The paper is very well written and illustrated and brings together

evidence from very diverse and excellent quality data sources. The methods and supplementary materials are also very clear. The work will be of interest to the readership of Nature Communications, as it proposes the 2022 Mauna Loa eruption was triggered by a 2-month-long sustained magmatic intrusion into a shallow reservoir (1-2 km beneath summit), which generated clear signs of eruption run-up that were stronger than previous, milder unrest linked to intrusions into the main reservoir (3-5 km beneath summit) over the last two decades. The work should have a positive impact on volcano monitoring efforts in Hawaii and globally. Below I provide some general comments for the authors' consideration, as well as minor line-by-line suggestions.

General comments

I particularly appreciated the thorough petrological characterisation of olivine zoning patterns, modelling of diffusion timescales, and fluid inclusion barometry. It is noteworthy that the team managed to obtain olivine separates from seemingly aphyric rocks by processing large volumes (>1 m³) of tephra samples to eventually obtain 'extremely rare olivine (n=64) and enstatite (n=2) macrocrysts > 0.5 mm in size' (L112-113) from four samples. The authors then exploit these crystals to provide quantitative constraints on magma history, storage depth and timescales, which is great. Why do you think there are so few large crystals? Is this common at Mauna Loa? The Fo content of the olivines is high: >Fo₈₇ cores and Fo₇₈₋₈₂ rims – the authors interpret the primitive olivines were transported from the primary reservoir (3-5 km) to the more evolved shallow reservoir (1-2 km) where they were partially dissolved (as informed by P zoning) and overgrown by low-Fo olivine. Do you think the rest of the primitive olivines were completely dissolved in the shallow reservoir? Was the shallow reservoir too evolved to form olivine cores? Would it have formed other crystals (only two opx found from this storage level) but they did not erupt?

The proposal of a shallow storage zone is very interesting. The authors describe 'The depth of the shallow reservoir at 1–2 km isn't apparent in earthquake catalogues' (L234) – why do you think that is? There are only few FI indicating shallow storage (6x FI in crystal rims/smaller crystals, in contrast with the main population of 31x FI that indicate storage at ca. 3.46 km, in the 1.8-4.1 km range) – Could there be one main storage zone (ca. 3-5 km), with the rim/microcryst-hosted FI recording entrapment during magma ascent to eruption (and that is why there are no low-Fo cores)? or are the olivine timescales too long for that (unless you consider much faster diffusion as suggested by Shea et al. (2023), but then there would be no correlations to geophysical monitoring datasets as the authors mention in the supplement)?

Line-by-line comments

L45: 'of both the'

L59: 'of both the'

L112: do you mean 'aphyric' instead of 'aphanitic'?

L115: the crystals come from four samples – could you provide details from these samples (e.g., coordinates, eruption date, tephra particle size) in the supplementary information?

L188: you may consider rewording to 'In Kilauea olivine, this texture has been...'

L121: I understand the depth of storage is constrained by FI barometry (not diffusion data)?

L288: 'of both the'

L531: 'broadly spaced'?

L542: 'minima'?

L614: 'the standard deviation of measured Fo contents results in an error of 17 K'?

L618: 'The magnitudes of these'

L743: I thought you had an upper and a lower (primary) reservoir but the latter one is described as intermediate here? Going back to Fig. 4 (which is a great summary!), I see you consider there may be storage at 13-14 km but this is not discussed – It may be good to clarify? Independently of that, the endnote is very nice.

I hope these comments are useful to the authors, congratulations on the excellent multi-disciplinary work! I really enjoyed going through the manuscript, thank you for the invitation to review.

With best wishes,
Teresa Ubide

Reviewer #1 (Remarks to the Author):

In this manuscript, Lynn et al. present the results of a multi-disciplinary study centred on the 2022 eruption of Mauna Loa on Hawaii. The authors combine mineral chemistry analysis with multiple geophysics datasets to better understand how to differentiate between non-eruptive

unrest and pre-eruptive unrest at the volcano. The authors track precursory activity in 2022 through mineral and melt chemistry, fluid inclusion barometry, numerical modelling of mineral zoning, syn-eruptive gas plume measurements, earthquake locations, seismic velocity changes, and ground deformation. They find evidence for a sustained intrusion of magma into a shallow magma reservoir only 1-2 km depth beneath the summit. The authors conclude that careful integration of geochemical and geophysical datasets is key for distinguishing non-eruptive from pre-eruptive unrest.

Overall, I think this is a really important piece of work that clearly demonstrates the vital importance of integrating geophysical and geochemical datasets. While I don't have enough experience to carefully evaluate the geochemical analyses presented here, I do appreciate that a huge amount of work has come together to produce this manuscript. I do not find many issues with the manuscript, and the issues I do find can be easily resolved; I have listed my comments below. Once these comments have been addressed, I look forward to seeing this work publishing in Nature Communications.

Oliver Lamb
Te Pū Ao | GNS Science
o.lamb@gns.cri.nz

Major comments

1. The authors emphasise in multiple places that their work can help distinguish between non-eruptive and pre-eruptive activity but I'm not entirely sure the difference has been made explicitly clear. For example, did the authors compare the geophysical measurements between 2022 and earlier unrest periods, particularly in 2014? In fig. 1B, the inflation measured across the MOKP-MPLS in the 2014 unrest is rapid but I could not see anything in the text that explains whether this deformation had come from a shallow or deep source. Similarly, where were the earthquakes in the 2014 event located? If there is a clear distinction between the 2014 unrest and the 2022 pre-eruptive unrest, this would help support their conclusions (especially on line 273 to 280).

The key difference in the 2022 events was the tilt recorded at MOK tilt meter, which we have noted is the first time we've seen this since its installation (Ellis et al. 2024, Ellis et al. preprint). We've added "Although MLSP-MOKP line length changes occurred in previous episodes of unrest (Fig. 1B), no corresponding MOK tilt was observed (Ellis et

al. preprint).” to Lines 167-169 to emphasize this difference. We have also added phrasing about the MOK tilt to Lines 316-319 (originally Lines 273-280 pointed out by the reviewer).

2. As a volcano–seismologist, I’ m left with some questions about the seismic data that could easily be answered:

> What is the cause of the earthquakes in NW flank, as they sit between the deeper magma storage at 13 km and the shallower ‘oma waena reservoir? I understand they originate from deep seated faults, as stated on line 147, but do you have an idea on what is causing them to rupture? Is it stress from ascending magma or from the deeper reservoir?

Yes, and our text in Lines 182-184 states this as the hypothesis. No additional change was made.

> Do you have an idea of why we see relatively little seismicity associated with the ‘oma luna reservoir, compared with the ‘oma waena reservoir? Is it because the shallow seismicity was dominated by tremor which might not be accounted for in earthquake catalogues, or was there simply no detectable seismicity in this region? If the latter, is there a distinct difference in the geology around each reservoir that might explain this difference in seismic activity?

A dedicated relocation effort would be required to say anything further with statistical significance related to shallow seismicity in this region. However, we know that the upper 2 km of the summit has some located seismicity (see for example Supplementary Figure S6). In September 2022, there is an increase in this shallow seismicity, but this increase is seen over all the relevant depths (again, see for example Figures S5 and S6). Tremor was not observed prior to the eruption. Because we cannot interpret shallow seismicity (or lack thereof) further, we have revised the text to remove language about the “lack of seismicity” and instead redirect the reader to the importance of the MOK shallow tilt signal and seismic velocities (see also responses to other related comments; Lines 324-327, 318-319).

Alternatively, is it possible that the use of a barometer parameterised for Kilauea is not appropriate for Mauna Loa resulting in estimated reservoir depths that are too shallow? I don’t really see this discussed anywhere in the manuscript. Please pardon my ignorance since I’ m not really experienced in geochemical barometry techniques!

Fundamentally, the fluid inclusion barometry method is extremely sensitive to the measured parameter, CO₂ density, and insensitive to the estimated parameters, temperature and XH₂O (for the range of values of these parameters in Hawaiian plumbing systems). More detail of this is found in DeVitre and Wieser (2024), and the method is discussed in detail in the Methods following the main text and supported by the Supplementary Material. We outline the differences between Kilauea and Mauna Loa temperature parameterization in Lines 640-644 and state that where the T-ranges

overlap with the Mauna Loa parameterization used for diffusion modeling there is good agreement between the two approaches. We also would not expect large differences in H₂O between Kilauea or Mauna Loa inclusions, based on similarities in measured melt inclusions (discussed in Lines 664-667, also shown in Figure S11). Furthermore, the propagated errors on the FI pressures/depths are generously accounted for (50-100 K temperature uncertainty allowed; Lines 690-694), and it can be seen how small the error bars are. Thus, the Kilauea parameterization is appropriate to apply.

Minor comments

Line 52 - 54: Do you have references you can cite for each of these examples?

References have been added

Line 62 - 63: I understand you've focused on diffusion chronometry studies because that's what this study is mostly focused on, but this seems a bit narrow. What about other studies that integrate geochemical with geophysical studies? For example, we did a study on Santiaguito volcano that integrated geophysical and geochemical measurements to shed light on why volcanic activity changed (Wallace et al., 2020). I would encourage the authors to expand the scope of their review of multi-disciplinary studies but I understand if this quickly becomes too much work to describe or reference.

We have added the phrase "... that leverage mineral chronometry" to the previous sentence in Lines 65-67 to emphasize that our focus is on those with mineral chronometers that can track magmas in time.

Fig. 1B: What do the ticks in the lower panel refer to? They don't seem to line up with the year labels on the x-axis.

Years and ticks were misaligned. Drawing error fixed in revised figure.

Line 102: I think you meant to refer to Figure 2A here, not Figure 1B.

Revision made.

Figure 3: From where on the volcano did the peaks seismicity 300-330 days prior to eruption come from? (panel C). There is also a typo in one of the labels here ('Increased siesmicity' , should be 'increased seismicity').

Spelling error corrected. The seismic peaks in question came from earlier NWF earthquake swarms with minor summit swarms that followed. I have added the NWF and Sum labels on these in panel C.

Line 157: I think 'correlates' would be a better word to use than 'corresponds' here.

Revision made.

Line 279: Partially related to one of my major comments, but how do we

detect magma ascending into the very shallowest portion of the reservoir system if there is no seismicity that comes from this region? Is it possible to detect this using the tilt signals alone? This is an important point to make as this kind of information might not be immediately available for volcano observatories to use.

We have revised this sentence (now Lines 324-327) to redirect the reader to focus on MOK tilt and velocity change (which mirrors the tilt): “The key geophysical monitoring signal distinguishing between unrest and the final run-up to eruption at Mauna Loa is therefore likely the change in MOK tilt¹¹ and seismic velocities signaling the migration of magma to the very shallowest portions of the reservoir system at 1–2 km depth beneath the summit.”

References

Wallace, P.A., Lamb, O.D., De Angelis, S., Kendrick, J.E., Hornby, A.J., Díaz-Moreno, A., González, P.J., von Aulock, F.W., Lamur, A., Utley, J.E.P., Rietbrock, A., Chigna, G., Lavallée, Y., 2020. Integrated constraints on explosive eruption intensification at Santiaguito dome complex, Guatemala. *Earth and Planetary Science Letters* 536, 116139. <https://doi.org/10.1016/j.epsl.2020.116139>

Reviewer #2 (Remarks to the Author):

Review of “Triggering an eruption at Mauna Loa, Earth’s largest volcano” by Lynn and colleagues

This ms presents a very detailed and comprehensive synthesis of multidisciplinary (both geochemical and geophysical) data associated with the most recent eruption (Nov 2022) at Mauna Loa, with the goal to advance the ability to forecast eruption. Such a multidisciplinary study is rare but critical for better understanding of eruption. It is important and of great interests to general readers. It should be published after moderate revision. I don’t have any major issue with the manuscript, and all my comments are to ask for clarification. I encourage the authors to make the manuscript more approachable to non-experts. Since I am a geochemist, my comments are related to mineral and melt inclusion chemistry. I leave the discussion on earthquakes to other reviewers.

1. Figure 3: Is timescale uncertainty of individual mineral considered in the histogram in Panel D? If possible, may I suggest replacing the histogram with a probability density function diagram, and taking into account the uncertainty of individual mineral?

The main representation of the timescale data in panel D is a cumulative distribution function with associated error on individual data points. We have removed the histogram from Figure 4, because it shows peaks and valleys that may not be real (subject to bin widths) and the cdf (main presentation of the data in panel D) is a better representation of timescale results. To avoid large amounts of white space in panel D, we have included a figure of a diffusion model fit instead (taken from previous Supplemental Figure 3C).

2. You may also want to make the X-axis of the inset the same as your main figure. This allows a direct comparison with other parameter.

The histogram has been removed – see responses to the above and below comments.

3. It is unclear how well (or bad) the diffusion time scales are correlated with other parameters. For example, there is a peak between 4–6 weeks. However, this peak seems not to be associated with any in Panels a–c.

As noted above, we have removed the histogram from Figure 4, because it shows peaks and valleys that may not be real (subject to bin widths) and the cdf (main presentation of the data) is a better representation of timescale results. We note that the uncertainty of individual results overlaps between the “apparent peaks” in the original histogram noted by the reviewer. Removal of the histogram will ensure that these are not overinterpreted.

4. Calculation of diffusion time scale:
a. A more detailed description is needed.

An additional paragraph of the numerical modeling setup has been added prior to the section on the Bayesian approach (see Lines 569-585).

b. Have the authors used a 2D or 3D diffusion model?

(see previous comment) New text in Lines 569-573 also state 1D with relevant equations.

c. The discussion about “olivine Fe–Mg diffusion coefficients” seems circular.

In response to another reviewer comment as well as this one, we added sentences to this Supplementary section that clarify the comparison. We want to acknowledge that recent literature has suggested diffusion coefficients need to be revisited, but based on geological constraints and our multiparametric comparisons, this seems not reasonable.

d. Some of the fitting may not reproduce the observation. For example, the first panel (ML22-ORI-1 o11) does not show good fitting at the corner.

This was due to the BayDiff using only the 1-sigma F_0 uncertainty (0.9 mol% F_0 ; Supplementary Data), when the 2-sigma uncertainty should have been used (0.18 mol% F_0). We re-ran BayDiff with the 2-sigma value and have replaced the figures in the Supplementary Information. These figures now show the models in better agreement

with the measured data, and the resulting timescale difference was negligible (Figure 3 revised accordingly, as well as Supplementary Data timescales dataset).

5. Figure 4: melt inclusion depth. I strongly suggest a more detailed description of how each individual melt inclusion depth is estimated, based on what parameter, and how (partial) degassing is excluded.

We would like to clarify that these are not melt inclusion data. They are fluid inclusion depths. All fluid inclusions trapped at Mauna Loa have experienced partial degassing, because degassing of CO₂ likely begins at ~30-50km depth in Hawai'i. However, what the fluid inclusions record is what the density of the degassing fluid was at the time they were trapped within a growing crystal, which tells us the pressure of the surrounding magma at the time of entrapment. Detailed descriptions of the measurements and calculations of fluid inclusion pressures and their corresponding depths are presented in both the methodology after the main text and in greater detail in the Supplementary Material.

6. Figure S8 and associated discussion: Again, a more detailed description is suggested, so that non-experts can understand.

We have added a paragraph of text in the Supplementary Material associated with Figure S8 that more broadly introduces fluid inclusions, their relevant literature, and the framework for understanding the model tests that were run.

7. Raw data are needed for the correlation of olivine core-rim composition and melt inclusion depth shown in Figure 4. They may be in Tables S4 and S7; however, I expect to see the above mentioned correlation better illustrated using data.

We revised Lines 258-265 to say *“The FI barometry is likely not influenced by significant re-equilibration (Supplementary Material; Fig. S8), and instead we interpret that they reflect the depths at which crystal growth is occurring”* which places the correlation between FI barometry to the process rather than the depth. We then try to link the inferences of the upper reservoir depth better with the other petrologic evidence in Lines 258-261: *“We suggest the reversely zoned enstatite macrocrysts with evolved core Mg# 78-79 grew at 1–2 km in equilibrium with the reservoir magmas in the upper reservoir and became reversely zoned up to Mg# 81-84 (out of equilibrium; Fig. S4) when mafic recharge magma intruded from the intermediate reservoir.*

An x-y plot looking for correlation of composition with depth is not practical for several reasons, including the relatively small size of the low-density FI dataset (n=6) and the fact that FIs are located at depth beneath the sectioned crystals (so as to preserve their fluids) so exact Fo contents for those FIs might be lower than inferred by projecting their position to the surface of the crystal. In order to get the x-y plot data implied by the reviewer, we would have to destroy the FIs.

For further support that low-density FIs are found (at depth below surface) near to crystal rims, please see Lynn_ML2022_SupplementaryInformation_Revision1.pdf ppt slides 42-44 (ML22-ORI-1_ol4) and slides 54-55 (ML22-ORI-1_ol7).

8. Finally, the reference styles are different in main text and appendix. Is numbered reference system required for both?

Supplementary references have been revised to follow the numbered formatting of the main text.

Reviewer #3 (Remarks to the Author):

This is a fantastic multi-disciplinary effort to combine petrological methods with geophysical monitoring (seismic and ground deformation data) and gas geochemistry to constrain the drivers of the 2022 Mauna Loa eruption. The paper is very well written and illustrated and brings together evidence from very diverse and excellent quality data sources. The methods and supplementary materials are also very clear. The work will be of interest to the readership of Nature Communications, as it proposes the 2022 Mauna Loa eruption was triggered by a 2-month-long sustained magmatic intrusion into a shallow reservoir (1-2 km beneath summit), which generated clear signs of eruption run-up that were stronger than previous, milder unrest linked to intrusions into the main reservoir (3-5 km beneath summit) over the last two decades. The work should have a positive impact on volcano monitoring efforts in Hawaii and globally. Below I provide some general comments for the authors' consideration, as well as minor line-by-line suggestions.

General comments

I particularly appreciated the thorough petrological characterisation of olivine zoning patterns, modelling of diffusion timescales, and fluid inclusion barometry. It is noteworthy that the team managed to obtain olivine separates from seemingly aphyric rocks by processing large volumes (>1 m³) of tephra samples to eventually obtain 'extremely rare olivine (n=64) and enstatite (n=2) macrocrysts > 0.5 mm in size' (L112-113) from four samples. The authors then exploit these crystals to provide quantitative constraints on magma history, storage depth and timescales, which is great.

Why do you think there are so few large crystals? Is this common at Mauna Loa?

Due to the paucity of mineral studies at Mauna Loa (noted in the main text and in the Supplementary Material), it's not possible at this time to determine if this is common for the recent 200 year period. We respond to the "why" posed by the reviewer in the response to the below comment.

The Fo content of the olivines is high: >Fo87 cores and Fo78 - 82 rims - the authors interpret the primitive olivines were transported from the primary reservoir (3-5 km) to the more evolved shallow reservoir (1-2 km) where they were partially dissolved (as informed by P zoning) and overgrown by low-Fo olivine. Do you think the rest of the primitive olivines were completely dissolved in the shallow reservoir? Was the shallow reservoir too evolved to form olivine cores? Would it have formed other crystals (only two opx found from this storage level) but they did not erupt?

These are excellent hypotheses and we have integrated this into our discussion in Lines 306-311. Ultimately more work is needed on Mauna Loa's reservoirs and their compositions.

The proposal of a shallow storage zone is very interesting. The authors describe 'The depth of the shallow reservoir at 1-2 km isn't apparent in earthquake catalogues' (L234) - why do you think that is?

See responses to Reviewer 1 above (similar comments about seismicity).

There are only few FI indicating shallow storage (6x FI in crystal rims/smaller crystals, in contrast with the main population of 31x FI that indicate storage at ca. 3.46 km, in the 1.8-4.1 km range) - Could there be one main storage zone (ca. 3-5 km), with the rim/microcryst-hosted FI recording entrapment during magma ascent to eruption (and that is why there are no low-Fo cores)? or are the olivine timescales too long for that (unless you consider much faster diffusion as suggested by Shea et al. (2023), but then there would be no correlations to geophysical monitoring datasets as the authors mention in the supplement)?

To best answer the questions posed by the author, more mineral chemistry studies of Mauna Loa eruption in the past 200 years are needed (see additional information in Supplementary Material). Within this study, the clearest evidence for requiring both the upper and the intermediate reservoirs is because of the contrasting chemistry between the "reservoir magmas" (and evolved erupted glass and whole rock compositions) of Rhodes (1988) and the presence of the high-Fo olivine cargo, which cannot live together. Reservoir magmas are mixed magmas that are "perched at the intersection of an olivine-control trend and a trend for more differentiated lava". The upper reservoir would be the more differentiated endmember, perhaps with little resident olivine. While alone, the fluid inclusion pressures could have been interpreted to have formed during ascent, or during shallow storage, formation during ascent is in conflict with the other evidence outlined above.

Line-by-line comments

L45: 'of both the'

Revision made.

L59: ‘of both the’

Revision made – “of both”

L112: do you mean ‘aphyric’ instead of ‘aphanitic’ ?

Revision made.

L115: the crystals come from four samples - could you provide details from these samples (e.g., coordinates, eruption date, tephra particle size) in the supplementary information?

Revision made. New section called “Mineral Sample Details” under the “Extended Methods” section of the Supplementary Material. Tephra particle size was not included because it wasn’t a measure/relevant parameter. In Line 138 I direct the reader to this table.

L188: you may consider rewording to ‘In Kīlauea olivine, this texture has been...’

Revision made.

L121: I understand the depth of storage is constrained by FI barometry (not diffusion data)?

Did the reviewer mean L221? If 221 (now Line 266), I have revised the start of the sentence by deleting the phrase “and diffusion”

L288: ‘of both the’

Revision made

L531: ‘broadly spaced’ ?

Revision made

L542: ‘minima’ ?

Revision made

L614: ‘the standard deviation of measured Fo contents results in an error of 17 K’ ?

Revision made

L618: ‘The magnitudes of these’

Revision made

L743: I thought you had an upper and a lower (primary) reservoir but the latter one is described as intermediate here? Going back to Fig. 4 (which is a great summary!), I see you consider there may be storage at 13-14 km but this is not discussed - It may be good to clarify? Independently of that, the endnote is very nice.

End note and ‘Ōlelo Hawai‘i removed due to other reviewer comments. Reservoirs have been termed upper and intermediate consistently throughout the revised text.

I hope these comments are useful to the authors, congratulations on the

excellent multi-disciplinary work! I really enjoyed going through the manuscript, thank you for the invitation to review.

With best wishes,
Teresa Ubide